# Outer membrane permeability of *Pseudomonas aeruginosa* through β-lactams: new evidence on the role of OprD and OpdP porins in antibiotic resistance

Francesco Amisano,[1] Paola Mercuri,[1] Steven Fanara,[2] Olivier Verlaine,[1] Patrick Motte,[2] Jean Marie Frère,[1] Marc Hanikenne,[2,3] Moreno Galleni[1]

**ABSTRACT** Gram-negative bacteria are a major concern for public health, particularly due to the continuous rise of antibiotic resistance. A major factor that helps the development of resistance is the outer membrane that is essential since it acts as a strong permeability barrier to many antibiotics that are effective against other bacteria. In this study, we determine the specific permeability coefficients for various antibiotics in *Pseudomonas aeruginosa* strains, which differ from each other for their porin expressions. We showed that OprD and OpdP porins contribute both to internalize meropenem and biapenem. Using qRT-PCR, we demonstrated that their expression is dependent of the various phases of cellular growth. We were able to show how the OpdP porin is less expressed in exponential growth phases, while it tends to be produced when the bacterial culture enters into the latent phase, in an inversely proportional way compared to the OprD porin. The deletion of the OpdP porin, in the presence of meropenem at concentrations equivalent to the MIC values, contributes to the selection of carbapenem-resistant strains. Therefore, the presence of mutations/deletions of the OpdP porin should receive greater consideration from a clinical point of view as the use of meropenem at nonoptimal concentrations could lead to the appearance of resistance phenotypes.

**IMPORTANCE** Carbapenem-resistant strains of *Pseudomonas aeruginosa* are among the major threats to public health. The permeability of the outer membrane for the β-lactam antibiotics is one of the major factors that reduce the activity of the antibiotics. In this study, we measure the low permeability coefficient of the *P. aeruginosa* outer membrane to β-lactams. The methodology we develop to determine the permeability can be applied to other antibiotic families and/or pathogens.

**KEYWORDS** OpdP porin, external membrane permeability, *Pseudomonas aeruginosa*, porins

Address correspondence to Moreno Galleni, mgalleni@uliege.be.

The authors declare no conflict of interest.

See the funding table on p. 20.

The gram-negative rod-shaped γ-proteobacterium *Pseudomonas aeruginosa* (*P. aeruginosa*) is a ubiquitous and opportunistic pathogen responsible for life-threatening infections, especially in immune-compromised patients, such as those with ventilator-associated pneumonia (VAP) or with urinary tract infections (UTIs) and is the leading cause of respiratory tract infections (RTIs) in cystic fibrosis patients (1–4). It is intrinsically resistant to different classes of antibiotics, and the few available therapeutical options include some β-lactam compounds, often delivered in combination with β-lactamase inhibitors, such as piperacillin/tazobactam, ceftazidime/avibactam, and imipenem/relebactam (5–8). Unfortunately, this bacterium has developed different mechanisms of resistance. In the case of β-lactam antibiotics, the major resistance

mechanism is the production of enzymes (β-lactamases) that can hydrolyze the β-lactam ring (3, 9). *P. aeruginosa* carries a chromosomally encoded and inducible AmpC β-lactamase (10). In addition, it can acquire extended-spectrum β-lactamases (ESBLs) and/or metallo-β-lactamases (MBLs), which confer resistance to carbapenems (11–13).

Moreover, *P. aeruginosa* is characterized by a low outer membrane permeability to β-lactams, due to the presence of OprF, a nonspecific pore, homologous to OmpC and OmpF from *Escherichia coli*. This porin is, indeed, present into two conformers, and the closed fraction represents approximately 95% of the total OprF expressed by the bacteria, leaving a minority in the open state, and its involvement in the permeation is still debated (14, 15).

As a consequence, hydrophilic nutrients and antibiotic internalization are mediated by a wide number of substrate selective channels (15).

Interplay with other resistance mechanisms, like the efflux pump upregulation, can extrude a wide variety of antibiotics present in the cytoplasmic or periplasmic space of the bacteria. In particular, increased expression of MexAB-OprM contributes to β-lactam resistance, as commonly found in clinical multiresistant isolates (16–18).

It is well documented that a decrease in the outer membrane permeability can be mediated by an altered expression of OprD. This porin facilitates basic amino acid uptake and influences antibiotic sensitivity, primarily to imipenem, due to the structural homology between arginine and the C2 antibiotic lateral chain (19, 20).

*P. aeruginosa* possesses 18 OprD homologs, characterized by their structural similarities and substrate specificities. They belong to the family of outer membrane carboxylate channel (Occ).

Within this family, two subgroups are identified: the OprD (or OccD) and the OpdK (or OccK) subfamilies (15, 21, 22).

OpdP (OccD3) shows the highest sequence identity with OprD (51%). It is associated with glycine–glutamate dipeptide translocation, and it has been assumed to be involved in meropenem uptake, although a clear phenotypic resistance profile in deletion mutants has not been determined (21, 23–26). This porin belongs to the *dppA4BCDF* operon, encoding the ABC machinery responsible for the utilization of dipeptides during the stationary phase, increasing the bacterial metabolic versatility. Its expression is controlled by the PsdR regulator (27, 28). Recent studies have shown that *psdR* is prone to acquire mutations, but their impact on the OpdP expression has not yet been described (29–31).

The research for antibiotic-specific channels has been directed to other porins belonging to the Occ family. For instance, in the study by Isabella and coworkers, porins selected for their expression quantified by RNAseq analysis after growth in minimal medium were hypothesized to be involved in the entry of antibiotics. The study identified the OpdC (OccD2), OpdT (OccD4), and OpdB (OccD7) porins as possible candidates. However, a *P. aeruginosa* isogenic mutant where the genes encoding three porins were deleted, together with *opdP* and *oprD,* did not display a modification of the antibiotic resistance profile compared to the single *oprD* mutant (24). Even a *P. aeruginosa* strain stripped of 40 porins resulted in MICs comparable to those of the single *oprD* knockout, suggesting the presence of alternative translocation pathways, independent of porins (32). For this reason, the MIC determination of *P. aeruginosa* isolates does not reflect the real pattern of porin expression and does not exhibit a reliable predictive value for bacterial permeability.

Therefore, an improved understanding of β-lactam translocation mechanisms in gram-negative bacteria might help in the design of new molecules, formulated also on the basis of their abilities to cross the outer membrane barrier.

Different methods have been proposed to study the outer membrane permeability in *P. aeruginosa*, starting from the pioneering work by Zimmermann and Rosselet (33). Comparing the periplasmic β-lactam hydrolysis of intact cells with the one obtained by a lysate made it possible to determine the outer membrane permeability coefficients for *P.*

*aeruginosa* (34). Unfortunately, this method turned out to be poorly reproducible due to the contribution of efflux pumps that interfere with antibiotic accumulation (35).

The use of radio-labeled β-lactams has been proposed as an alternative method for the quantification of the antibiotic periplasmic concentration (36). However, this method is limited by the difficulty of obtaining a wide set of radioactive compounds.

Whole-cell analytical techniques, including mass spectrometry-based analysis, allow the quantification of the variation in extracellular antibiotics or the measurement of the direct accumulation in the periplasm (37–39), but they are highly time-consuming.

Another approach being pursued was the study of single porin permeation properties using different techniques such as the liposome swelling assay, electrophysiology, or molecular dynamics simulations. They have importantly contributed to the definition of the specific role of single porins, the determination of their specific conductance, and have identified the consequences that mutations may have on translocation properties (40–44). Nevertheless, the study of the single porin properties undoubtedly failed to comprehend the complexity of the bacterial response to antibiotics, given that synergic effects are not noticeable.

In a first approach, we determined the *P. aeruginosa* outer membrane permeability toward β-lactam, as described below. This method exploits the property of BlaR-CTD, a soluble penicillin-binding protein that displays a high affinity for β-lactams. Its expression in the periplasm allows an accurate estimation of the quantity of the antibiotic that permeates through the outer membrane (45).

Unlike the previously mentioned methods, the study of single or multiple porin(s) isogenic mutants can elucidate the role of a single porin or the presence of any synergic effect in double or multiple knockout strains for a broad variety of β-lactams, and this was the first aim of our research.

The second goal of this study was to ascertain the real contribution of OpdP in carbapenem resistance, especially under stress conditions. To the best of our knowledge, we were the first to investigate the expression of different porins during different bacterial growth phases by means of qRT-PCR, and we believe that this is a useful tool to broaden our understanding of the response to antibiotic therapy.

Our third goal was to verify whether the single deletion of the OpdP porin confers a selective advantage in developing a phenotype of carbapenem resistance. To this end, we performed a multistep resistance experiment using meropenem at sub-minimum inhibitory concentrations and subsequently analyzed the resistant mutants thus obtained.

Finally, we performed whole-genome sequencing on selected strains to clarify the specific resistance genotype.

## RESULTS AND DISCUSSION

### Antibiotic resistance determination and mutant selection

Different subclasses of β-lactams were tested to evaluate the MIC variations associated to porin deletions, and the resistance profiles of *P. aeruginosa* strains are reported in Table 1.

MIC values for carbapenems were increased in all the strains where OprD was deleted and also in TNP004, described to downregulate OprD expression (47). The deletions of the other porins did not change the resistance phenotype and did not suggest any synergic effect for the antibiotic tested.

We further obtained *P. aeruginosa* mutant strains, derived from *P. aeruginosa* PAO1 and *P. aeruginosa* ARC5170 (PAO1Δ*opdP*), with the help of a multistep resistance experiment.

Colonies were cultured and then plated at different growth phases at sub-MIC meropenem concentrations. We registered the appearance of resistant colonies for both strains and at the different growth phases tested, but ARC5170 proved to be the most adept at acquiring the ability to grow in the presence of meropenem, 30 times more frequently than for the other strain; in particular, we found approximatily 600 colonies derived from ARC5170, while only 20 were derived from PAO1.

**TABLE 1** MIC values for the different *P. aeruginosa* strains[a,b]

| Antibiotics (µg/mL) | CLSI Standard | CLSI Susc. | *P. aeruginosa* MICs (µg/mL) | | | | | | | | | | | | | | | |
|---|---|---|---|---|---|---|---|---|---|---|---|---|---|---|---|---|---|---|
| | | | ATCC 27853 | PAO1 | PAO1-Jap | TNP004 | ARC545 | ARC5990 (Δoprd) | ARC5170 (Δopdp) | ARC5782 (Δoprd, ΔopdP) | ARC5998 (Δ5porins) | LG01 | LG02 | LG03 | LG04 | LG05 | LG06 | LG07 |
| Ampicillin | NA | NA | 2,000 | 2,000 | 2,000 | 1,000 | 2,000 | 1,000 | 1,000 | 1,000 | 1,000 | 2,000 | 2,000 | 2,000 | ND | ND | ND | ND |
| Benzylpenicillin | NA | NA | >2,000 | >2,000 | >2,000 | >2,000 | >2,000 | >2,000 | >2,000 | >2,000 | >2,000 | >2,000 | >2,000 | >2,000 | ND | ND | ND | ND |
| Piperacillin | 1–8 | ≤16 | 2 | 2 | 2 | 2 | 2 | 2 | 2 | 2 | 2 | 8 | 4 | 8 | ND | ND | ND | ND |
| Cefalotin | NA | NA | >2,000 | >2,000 | >2,000 | >2,000 | >2,000 | >2,000 | >2,000 | >2,000 | >2,000 | ND | ND | ND | ND | ND | ND | ND |
| Cephaloridine | NA | NA | >2,000 | >2,000 | >2,000 | >2,000 | >2,000 | >2,000 | >2,000 | >2,000 | >2,000 | ND | ND | ND | ND | ND | ND | ND |
| Cefoxitin | NA | NA | 1,000 | 1,000 | 1,000 | 500 | 1,000 | 1,000 | 1,000 | 1,000 | 1,000 | ND | ND | ND | ND | ND | ND | ND |
| Cefuroxime | NA | NA | 250 | 250 | 250 | 250 | 250 | 500 | 500 | 500 | 500 | >500 | >500 | >500 | ND | ND | ND | ND |
| Cefotaxime | 8–32 | NA | 16 | 16 | 16 | 8 | 16 | 16 | 16 | 16 | 16 | 32 | 16 | 32 | 16 | 16 | 16 | 16 |
| Ceftazidime | 1–4 | ≤8 | 1 | 1 | 1 | 2 | 1 | 1 | 1 | 1 | 1 | 2 | 1 | 2 | ND | ND | ND | ND |
| Cefepime | 0.5–4 | ≤8 | 1 | 1 | 1 | 0.5 | 1 | 1 | 1 | 1 | 1 | 2 | 2 | 2 | ND | ND | ND | ND |
| Imipenem | 1–4 | ≤2 | 1 | 1 | 1 | **8** | 1 | **8** | 1 | **8** | **8** | **16** | **16** | **8** | **16** | **16** | **16** | **16** |
| Meropenem | 0.12–1 | ≤2 | 0.5 | 0.5 | 0.5 | **2** | 0.5 | **4** | 0.5 | **4** | **4** | **8** | **4** | **8** | **4** | **4** | **4** | **4** |
| Ertapenem | 2–8 | NA | 8 | 8 | 8 | **32** | 8 | **32** | 8 | **32** | **32** | **64** | **64** | **64** | **64** | **64** | **64** | **64** |
| Biapenem | 0.5–2 | NA | 0.5 | 0.5 | 0.5 | **4** | 0.5 | **4** | 0.5 | **4** | **4** | **4** | **4** | **4** | **4** | **4** | **4** | **4** |
| Doripenem | 0.12–0.5 | ≤2 | 0.25 | 0.25 | 0.25 | **1** | 0.25 | **1** | 0.25 | **1** | **1** | **2** | **2** | **2** | **2** | **2** | **2** | **2** |
| Tetracycline | 8–32 | ≤4 | 8 | 8 | 8 | 8 | 8 | 8 | 8 | 8 | 8 | 16 | 8 | 16 | ND | ND | ND | ND |
| Gentamicin | 0.5–2 | ≤4 | 1 | 1 | 1 | 2 | 1 | 2 | 2 | 2 | 2 | ND | ND | ND | ND | ND | ND | ND |

[a]CLSI standard refers to acceptable limits for quality control strains used to monitor the accuracy of MICs. CLSI susc. refers to the MIC susceptibility breakpoints interpreted by CLSI (46). NA: not available, ND not determined.
[b]The bold characters represent the names of the *Pseudomonas* strains utilized in this study.

We selected *P. aeruginosa* LG01, derived from PAO1, and six mutants derived from ARC5170 named *P. aeruginosa* LG02-LG07; determined their MICs; and observed that all the strains presented a stable resistance profile characterized by a reduced sensitivity to carbapenems, identical to that of the strains deprived of OprD (Table 1).

## OprD sequencing and whole-genome sequencing

*P. aeruginosa* TNP004 was described to produce an undetectable amount of OprD (47), but the cause underlying this downregulation has not been elucidated yet. For this reason, we sequenced the *oprD* gene and observed that it contained a single-nucleotide mutation (T1301C) yielding the Leu434Pro mutation in OprD, a modification in a transmembrane domain that increases the instability of the porin (GenBank accession number OR069747).

We were also interested in assessing whether the carbapenem resistance found in the strains selected under meropenem pressure was attributable to *oprD* mutations and consequently performed Sanger sequencing. For *P. aeruginosa* LG01, the insertion of one cytosine between positions 1,205 and 1,206 was found to induce a frameshift in the *oprD* open reading frame (GenBank accession number OR069748). LG02 is characterized by a single-nucleotide substitution (G1017A), introducing a premature STOP codon (GenBank accession number OR069749). The strains LG04, LG05, LG06, and LG07 shared the same nucleotide deletion in position 1,291 (G), resulting in a frameshift and in the synthesis of a truncated porin (GenBank accession number OR069750). Figure S3 summarizes the alignments of the different *oprD* mutated genes. Interestingly, *P. aeruginosa* LG03 did not exhibit any mutation when compared to the *oprD* wild-type gene, despite a carbapenem resistance profile similar to that of the other *oprD* mutants.

The whole-genome sequences of *P. aeruginosa* TNP004 and LG03 were determined to identify other mutations that could lead to the carbapenem resistance genotype associated to those strains. Due to the report of many polymorphisms in different PAO1 reference strains diffused worldwide (48), PAO1-Jap and ARC545, parental strains of TNP004 and ARC5170, respectively, were sequenced in order to exclude the role of mutations already present in these strains for carbapenem resistance observed in TNP004 and LG03.

The nucleotide sequences were deposited in the GenBank database, Bioproject PRJNA985251 under accession number SAMN35794375-8, while the observed mutations are reported in Fig. 1.

TNP004, besides the nucleotide substitution in *oprD*, presented other mutations including a deletion in *fliF* and two point mutations in *lasR* and *pilR*. However, these genes have not been described to be involved in the regulation of OprD expression, and so we concluded that the mere amino acid substitution in OprD causes the lack of porin's expression.

We also observed that, when compared to the reference PAO1, LG03 exhibited two single-nucleotide polymorphisms in *nalD* and *dsbS* in addition to the *opdP* deletion.

*dsbS* was recently described as a histidine-kinase sensor that acts together with the cognate response regulator *dsbR* in copper homeostasis (49). The copper-induced response has been previously shown to reduce the OprD expression through a regulation mediated by different two-component systems, such as *czcRS* and *copRS* (50, 51). It is reasonable to speculate that also this third copper regulation system might govern OprD expression. *nalD* is a transcriptional repressor of cellular efflux whose mutations have been correlated with MexAB-OprM overexpression (52); the upregulation of another efflux pump system, MexEF-OprN, mediated by the regulator *mexT* is known to decrease OprD expression (53). However, although many strains have been described to exhibit mutations in *nalD* causing MexAB-OprM overexpression, there is no evidence in the literature that these mutations could directly influence OprD.

The sequencing has therefore revealed that a single mutation can cause carbapenem resistance in TNP004, while *oprD* mutations for six of the seven mutants selected under meropenem pressure alter the correct OprD synthesis, giving rise to increased resistance

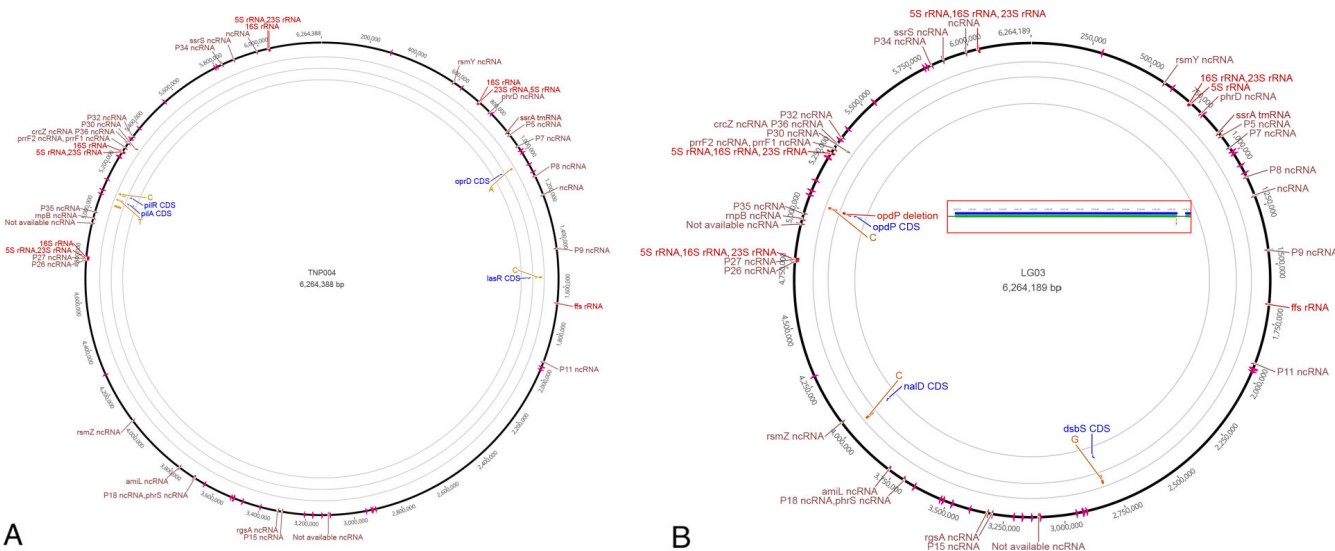

**FIG 1** Circular visualization of the genome assembly obtained using the software Geneious (version R10). (A) The genome sequence of TNP004 was compared to that of its parental strain PAO1-Jap. The mutations between the genomes are shown. (B) Genome sequence of LG03 was compared with that of its parental strain ARC545. The opdP deletion is highlighted, and the other observed mutations are shown.

to carbapenems. Furthermore, the whole-genome sequencing has revealed the presence in LG03 of two mutations that contribute to carbapenem resistance, but their actual contribution requires further investigations.

## Western blot

The expression of OprD in different mutant strains was verified by means of Western Blot (Fig. 2). As assumed, we confirmed the absence of the porin not only in the deleted mutants but also in the *oprD* mutant TNP004 and in the LG01, LG02, and LG04-LG07 strains, confirming the sequencing results. Interestingly, we could observe that LG03 was characterized by a downregulation of OprD, thus resulting in a consistent resistance profile to carbapenems.

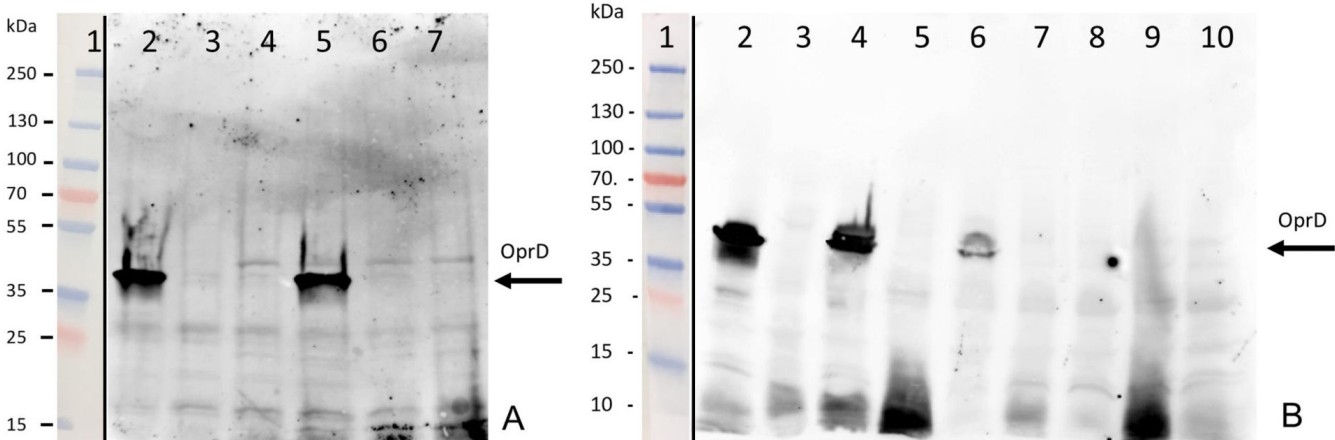

**FIG 2** OprD detection performed by Western Blot in the following strains: (A). 1, protein marker; **2**,PAO1; 3,TNP004; **4**, ARC5990 (PAO1.ΔoprD); **5**, ARC5170 (PAO1ΔopdP); **6**, ARC5782 (PAO1ΔoprD, ΔopdP) and **7**, ARC5998 (PAO1ΔoprD, ΔopdP,ΔopaB, ΔopdC, and ΔopdT). (B). **1**, protein marker; **2**, PAO1; 3, LG01; **4**, ARC5170 (PAO1ΔopdP); **5**, LG02; **6**, LG03; **7**, LG04; **8**, LG05; **9**, LG06 and 10, LG07. The porin OprD is marked by an arrow. Protein markers' pictures were put aside the chemiluminescence scans.

## Growth curves

The planktonic growth in the LB medium of different porin mutants was compared to that of the wild-type PAO1 (Fig. 3). Remarkably, we did not observe an important delay in the growth of the different tested mutants compared to *P. aeruginosa* PAO1. Even ARC5998, with five different porins deleted, was able to grow in the LB medium at a rate similar to that of the PAO1 wild-type strain.

A possibly relevant difference is shown only in TNP004 where the cell density is slightly larger compared to the other strains. We could so assess that the lack of one or more porins does not influence bacterial growth, at least in the LB medium.

## Permeability coefficient determination

A better understanding of the permeability of gram-negative bacteria, and in particular of *P. aeruginosa*, is an important factor for better directing the search for new antibiotics (54, 55).

While the role of the OprD porin in imipenem resistance is now widely understood (20, 56), the permeation of other antibiotics through the outer membrane remains difficult to interpret (15, 26, 57); indeed, antibiotic diffusion due to other porins does not seem to be sufficient to explain their uptake (32).

With this aim in mind, we adapted a previously described protocol (45) to quantify the β-lactam translocation into *P. aeruginosa* periplasm in order to determine the permeability coefficients of the outer membrane to different β-lactams. We performed validations to exclude major interfering events that could occur during measurements that include the following: i) the absence of MIC variation as a consequence of BlaR-CTD periplasmic production, ii) the verification of BlaR-CTD high affinity for the β-lactams tested, and iii) the absence of significant AmpC induction in the presence of β-lactams.

We transformed *P. aeruginosa* PAO1, ARC5990 (PAO1Δ*oprD*), ARC5170 (PAO1Δ*opdP*), ARC5782 (PAO1Δ*oprD*, Δ*opdP*), ARC5998 (PAO1 five porins mutant) with plasmid pKT240blaR, and TNP004 (PAO1 ↓OprD) with pKT240blaR-gen, a derived plasmid carrying gentamicin resistance, to produce BlaR-CTD in the periplasm of these strains.

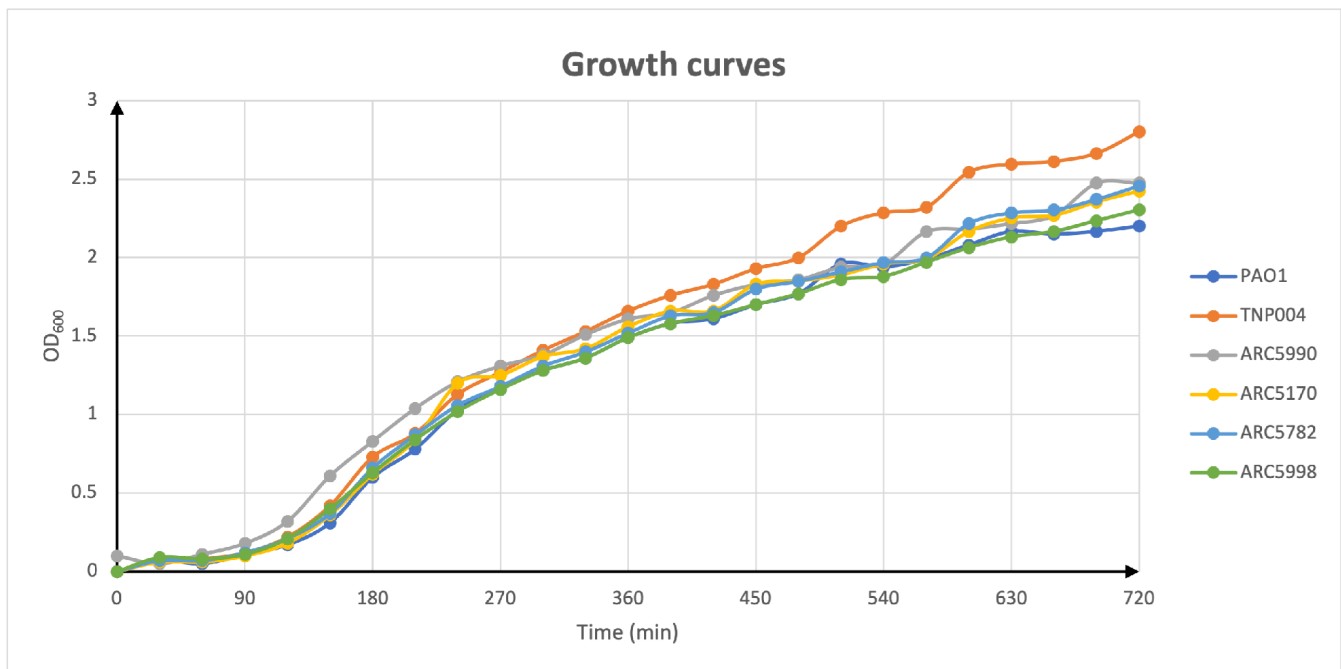

**FIG 3** Growth curves of *P. aeruginosa* strains in LB cultures.

We compared the MIC profiles of the strains producing BlaR-CTD or not. We also transformed PAO1 with an empty vector (pKT240neg) as a control to compare the effects induced by the plasmid alone to those caused by BlaR-CTD expression.

As reported in Table 2, the presence of BlaR in the periplasmic space did not alter the MIC values of the host bacteria with the exception of three compounds, piperacillin, ceftazidime, and cefepime, that presented unexpected MIC increases ranging from twofold to fourfold dilutions when compared to the *blaR*-devoid strain.

This phenomenon could be explained by a lower affinity of these three antibiotics for natural PBPs compared to the other antibiotics tested. A similar conclusion was made by Montaner and coworkers studying the effect of AmpC hyperexpression (58). The periplasmic BlaR-CTD production, similarly to AmpC hydrolysis, might reduce the intracellular concentration of these three antibiotics to a limit level that would change the PBP occupancy causing increases in MICs, or, equivalently, might induce AmpC expression. However, whatever the exact cause, we excluded these antibiotics from our permeability study and preferred to concentrate on antibiotics whose MICs were not altered by BlaR-CTD expression.

We successively measured the acylation constants of BlaR-CTD with the different β-lactam antibiotics (Table 3). Our data confirmed that the formation of a stable acyl-enzyme was not a rate-limiting step in our experiments ($k_2/K' > 0.02\ \mu M^{-1}\ s^{-1}$).

We finally verified that the expression of the chromosome-encoded AmpC β-lactamase did not affect the detection of labeled BlaR-CTD. To do so, we determined the level of production of the AmpC β-lactamase in the different *P. aeruginosa* PAO1 cultures. The concentration of the antibiotic added to the culture was identical to its maximum concentration used during the permeability assay, and the incubation times for different β-lactams were equal to the maximum duration of the permeability measurements for the specific antibiotic. We, therefore, determined the AmpC concentration in the function of the incubation time (Table 4).

We showed that the specific activity of the chromosome-encoded AmpC remained similar to those obtained for the negative controls. Nevertheless, ertapenem showed a minor increase in the AmpC activity (fourfold compared to the negative control). The validity of the assay to detect increased expression of the β-lactamase was demonstrated by verifying the strong AmpC induction after a longer incubation (6 hours) in the presence of ampicillin and cefoxitin, known to be good AmpC inductors (59).

Antibiotic penetration was assessed in planktonic cultures grown in LB, and the assay was performed during the late exponential growth phase ($A_{600}$ approximatively 1.6), thus excluding an increase in periplasmic BlaR-CTD due to bacterial duplication during the short duration of the analysis (maximum 40 minutes).

We first determined the permeability coefficients for different β-lactams in *P. aeruginosa* PAO1, and the results are reported in Table 5 (60–62). One can immediately notice the lower outer membrane permeability of *P. aeruginosa* when compared to that of *E. coli*.

We then selected a penicillin (ampicillin), a first-generation cephalosporin (cephaloridine) and five carbapenems (imipenem, meropenem, ertapenem, doripenem, and biapenem) and determined their permeability coefficients in different mutant strains (Table 6). As expected, decreased values for imipenem uptake were detected in all the strains where OprD was deleted or mutated. The OprD mutants exhibited a 130-fold reduction in the permeability coefficient when compared to PAO1. The values obtained for the TNP004 mutant are similar to those for ARC5990 (PAO1Δ*oprD*), confirming that the porin is not expressed in TNP004.

This result is in accordance with the specific role of OprD in imipenem uptake, due to the structural identity between the C2 side-chain of imipenem and that of arginine, the natural substrate of the porin. For the other tested carbapenems, the increase in MIC values does not clearly map into a marked permeability decrease since we only measured a twofold difference between the single OprD mutants and the wild-type.

**TABLE 2** MIC values of the different *P. aeruginosa* strains producing or not BlaR-CTD[a,b]

| Antibiotics (µg/mL) | CLSI Standard | CLSI Susc. | PAO1 | PAO1 pKT240neg | PAO1 pKT240blaR | TNP004 | TNP004 pKT240blaR | ARC5990 (Δoprd) | ARC5990 pKT240blaR (Δoprd) | ARC5170 (Δopdp) | ARC5170 pKT240blaR (Δopdp) | ARC5782 (Δoprd, ΔopdP) | ARC5782 pKT240blaR (Δoprd, ΔopdP) | ARC5998 (Δ5porins) | ARC5998 pKT240blaR (Δ5porins) |
|---|---|---|---|---|---|---|---|---|---|---|---|---|---|---|---|
| Ampicillin | NA | NA | 2,000 | 1,000 | 1,000 | 1,000 | 500 | 1,000 | 1,000 | 1,000 | 1,000 | 1,000 | 1,000 | 1,000 | 1,000 |
| Benzylpenicillin | NA | NA | >2,000 | >2,000 | 2,000 | >2,000 | >2,000 | >2,000 | >2,000 | >2,000 | >2,000 | >2,000 | >2,000 | >2,000 | >2,000 |
| Piperacillin | 1–8 | ≤16 | 2 | 1 | 4 | 2 | 4 | 2 | **8** | 2 | **8** | 2 | **8** | 2 | **8** |
| Cefalotin | NA | NA | >2,000 | >2,000 | >2,000 | >2,000 | 2,000 | >2,000 | >2,000 | >2,000 | >2,000 | >2,000 | >2,000 | >2,000 | >2,000 |
| Cephaloridine | NA | NA | >2,000 | >2,000 | >2,000 | >2,000 | 2,000 | >2,000 | >2,000 | >2,000 | >2,000 | >2,000 | >2,000 | >2,000 | >2,000 |
| Cefoxitin | NA | NA | 1,000 | 1,000 | 500 | 500 | 500 | 1,000 | 1,000 | 1,000 | 1,000 | 1,000 | 1,000 | 1,000 | 1,000 |
| Cefuroxime | NA | NA | 250 | 250 | 250 | 250 | 250 | 500 | 500 | 500 | 500 | 500 | 500 | 500 | 500 |
| Cefotaxime | 8–32 | NA | 16 | 16 | 16 | 8 | 16 | 16 | 16 | 16 | 16 | 16 | 16 | 16 | 16 |
| Ceftazidime | 1–4 | ≤8 | 1 | 1 | **16** | 2 | **32** | 1 | **16** | 1 | **16** | 1 | **16** | 1 | **16** |
| Cefepime | 0.5–4 | ≤8 | 1 | 1 | **4** | 1 | 4 | 1 | 4 | 1 | 4 | 1 | 4 | 1 | 4 |
| Imipenem | 1–4 | ≤2 | 1 | 1 | 1 | 8 | 8 | 8 | 8 | 1 | 1 | 8 | 8 | 8 | 8 |
| Meropenem | 0.12–1 | ≤2 | 0.5 | 0.5 | 0.5 | 2 | 2 | 4 | 4 | 0.5 | 0.5 | 4 | 4 | 4 | 4 |
| Ertapenem | 2–8 | NA | 8 | 8 | 8 | 32 | 32 | 32 | 32 | 8 | 8 | 32 | 32 | 32 | 32 |
| Biapenem | 0.5–2 | NA | 0.5 | 0.5 | 0.5 | 4 | 4 | 4 | 4 | 0.5 | 0.5 | 4 | 4 | 4 | 4 |
| Doripenem | 0.12–0.5 | ≤2 | 0.25 | 0.25 | 0.25 | 1 | 1 | 1 | 1 | 0.25 | 0.25 | 1 | 1 | 1 | 1 |
| Tetracycline | 8–32 | ≤4 | 8 | >128 | >128 | 8 | >128 | 8 | >128 | 8 | >128 | 8 | >128 | 8 | >128 |
| Gentamicin | 0.5–2 | ≤4 | 1 | 1 | 2 | 2 | 16 | 2 | 2 | 2 | 2 | 2 | 2 | 2 | 2 |

[a]The values previously obtained for the non-transformed strains are reported. The CLSI standard refers to acceptable limits for quality control strains used to monitor the accuracy of MICs. CLSI susc. refers to the MIC susceptibility breakpoints interpreted by CLSI (46). NA: not available, ND not determined.

[b]The bold characters represent the names of the Pseudomonas strains utilized in this study.

**TABLE 3** Acylation rate constant ($K_2/K$) for the different tested antibiotics

| Antibiotic | $k_2/K$ ($\mu M^{-1} \cdot s^{-1}$) | Reference |
|---|---|---|
| Nitrocefin | 3.6 ± 0.3 | This study |
| Benzylpenicillin | 8.7 ± 1.1 | (57) |
| Ampicillin | 1.3 ± 0.1 | (57) |
| Cephaloridine | 5.9 ± 0.2 | (57) |
| Cefoxitin | 0.06 ± 0.02 | (57) |
| Cefuroxime | 0.02 ± 0.005 | (57) |
| Cefotaxime | 0.04 ± 0.003 | (57) |
| Imipenem | 0.8 ± 0.2 | This study |
| Meropenem | 0.8 ± 0.2 | This study |
| Ertapenem | 1.1 ± 0.2 | This study |
| Biapenem | 1.4 ± 0.3 | This study |
| Doripenem | 1.7 ± 0.3 | This study |

The permeability coefficients of ARC5170 (PAO1Δ*opdP*) were similar to those of the reference strain PAO1, suggesting that OpdP was not involved in antibiotic uptake.

Interestingly, a different result was noticed in the analysis of the mutant ARC5782, that lacks both OprD and OpdP. The imipenem permeability coefficient was similar to that of ARC5990 (PAO1Δ*oprD*), while, for meropenem and biapenem, the permeability coefficients were respectively tenfold and 30-fold lower than that of the wild-type PAO1.

These data underline a synergistic role of the OprD and OpdP porins in meropenem and biapenem uptake. An involvement of OpdP in meropenem permeation had already been suggested in the literature (24), but the diffusion of biapenem through this porin had never been described before.

We performed the same analysis on ARC5998 and obtained data similar to those observed with the double OprD and OpdP mutants, indicating that OpdB, OpdC, and OpdT are not primarily involved in the uptake of the tested antibiotics.

This approach allowed the evaluation of the specific permeability coefficients for various antibiotics in a series of *P. aeruginosa* strains (Table 6), which exhibit different levels of porin expressions.

**TABLE 4** Periplasmic AmpC concentration and specific activity of the AmpC β-lactamase in the presence or absence (/) of β-lactams for the different *P. aeruginosa* cultures[a]

| Incubation time (min) | Antibiotic (final concentration µM) | Periplasmic AmpC (mg/L) | Specific activity ($\mu mol \cdot min^{-1} \cdot mg^{-1}$) |
|---|---|---|---|
| 12 | Imipenem (0.02) | 0.029 | 0.07 |
| | Doripenem (1) | 0.024 | 0.06 |
| | Biapenem (0.04) | 0.020 | 0.05 |
| | / | 0.026 | 0.06 |
| 20 | Meropenem (2) | 0.020 | 0.05 |
| | / | 0.042 | 0.10 |
| 30 | Ampicillin (20) | 0.026 | 0.05 |
| | Benzylpenicillin (40) | 0.027 | 0.08 |
| | Cephaloridine (7.5) | 0.024 | 0.08 |
| | Ertapenem (7.5) | 0.073 | 0.25 |
| | / | 0.029 | 0.06 |
| 40 | Cefoxitin (30) | 0.49 | 1.25 |
| | Cefuroxime (60) | 0.018 | 0.03 |
| | Cefotaxime (60) | 0.021 | 0.03 |
| | / | 0.017 | 0.03 |
| 360 | Ampicillin (50) | 22.6 | 1.43 |
| | Cefoxitin (30) | 90.4 | 5.76 |
| | / | 4.88 | 0.31 |

[a]The activity was followed by nitrocefin hydrolysis. (/) refers to a negative culture where no antibiotic was added.

**TABLE 5** Permeability coefficients determined in *P. aeruginosa* PAO1 for a set of different β-lactams, belonging to penicillin, cephalosporin (1st, 2nd, and 3rd generation), and carbapenem families[a]

| | Permeability coefficients (nm/sec) | |
|---|---|---|
| Antibiotic | *P. aeruginosa* PAO1 | *E. coli* |
| Ampicillin | 0.008 ± 0.004 | 28 (70) |
| Benzylpenicillin | 0.006 ± 0.002 | |
| Cephaloridine | 0.03 ± 0.01 | |
| Cefoxitin | 0.002 ± 0.0006 | |
| Cefuroxime | 0.001 ± 0.0004 | |
| Cefotaxime | 0.001 ± 0.0005 | 180 (72) |
| Imipenem | 20 ± 9 | 1800 (71) |
| Meropenem | 0.06 ± 0.01 | 300 (71) |
| Ertapenem | 0.06 ± 0.02 | |
| Doripenem | 0.56 ± 0.38 | |
| Biapenem | 4.7 ± 1.4 | |

[a]Values reported in the literature for *E. coli* are shown for comparison.

We also highlighted the presence of compensatory effects in the permeation of antibiotics, in particular, the contribution of both OprD and OpdP porins in the internalization of meropenem and biapenem. Deleting each of these porins individually does not alter the entry rate of these antibiotics in the respective mutants, while in the mutant deprived of both porins, a marked decrease in permeability for both antibiotics is observed (tenfold and 30-fold, respectively). This outcome could not have been inferred from the MIC values, which are in fact identical in all mutants for the antibiotics mentioned above (Table 1).

The method described here can, in principle, be extended to other antibiotics and also to other gram-negative pathogens of clinical interest, representing a useful tool to study porin's functionality.

## qRT-PCR

The relative expression of *oprD*, together with four other porins (*opdP*, *opdB*, *opdC*, and *opdT*) at four different time points of cellular growth was quantified by qRT-PCR. Total mRNAs were so extracted at $OD_{600}$ of 0.6, 1.2, 1.6, and 2.0, corresponding, respectively, to the early, mid and late exponential, and early stationary phase.

*PA3340*, *gyrA,* and *cysG* genes, due to their relative stable expressions, were chosen as reference genes, and the exact protocol that results in their selection is reported in the Supplementary Material.

The relative *oprD* expression in *P. aeruginosa* PAO1, TNP004, and ARC5170 (PAO1*ΔopdP*) is reported in Fig. 4A. We confirmed that, as previously assumed, the expression of *oprD* mRNA is inversely proportional to cell density (53); as a consequence, the decreased expression of *oprD* observed at $OD_{600}$ 1.6, compared to the early

**TABLE 6** Permeability coefficients for different β-lactams determined for *P. aeruginosa* PAO1 and other porin(s) or efflux pump mutants[a]

| | Permeability coefficients (nm/sec) | | | | | |
|---|---|---|---|---|---|---|
| Antibiotic Relevant characteristics | PAO1 | TNP004 ↓OprD | ARC5990 *ΔoprD* | ARC5170 *ΔopdP* | ARC5782 *ΔoprD, ΔopdP* | ARC5998 *ΔoprD, ΔopdP, ΔopdB, ΔopdC,* and *ΔopdT* |
| Ampicillin | 0.008 ± 0.005 | 0.008 ± 0.003 | 0.02 ± 0.01 | 0.01 ± 0.002 | 0.01 ± 0.002 | 0.02 ± 0.005 |
| Cephaloridine | 0.03 ± 0.02 | 0.02 ± 0.004 | ND | 0.03 ± 0.01 | 0.03 ± 0.01 | 0.04 ± 0.01 |
| Imipenem | 20 ± 9 | 0.13 ± 0.07 | 0.14 ± 0.07 | 15 ± 5.8 | 0.13 ± 0.05 | 0.12 ± 0.06 |
| Meropenem | 0.06 ± 0.01 | 0.03 ± 0.02 | 0.03 ± 0.01 | 0.1 ± 0.05 | 0.006 ± 0.002 | 0.01 ± 0.005 |
| Ertapenem | 0.06 ± 0.02 | 0.03 ± 0.01 | 0.04 ± 0.02 | 0.02 ± 0.01 | 0.02 ± 0.01 | 0.02 ± 0.01 |
| Doripenem | 0.51 ± 0.35 | 0.16 ± 0.06 | 0.07 ± 0.03 | 0.11 ± 0.02 | 0.14 ± 0.11 | 0.11 ± 0.05 |
| Biapenem | 4.7 ± 1.4 | 3.4 ± 1.9 | 4.2 ± 2.7 | 7.2 ± 2.9 | 0.21 ± 0.13 | 0.12 ± 0.03 |

[a]ND refers to a coefficient not determined.

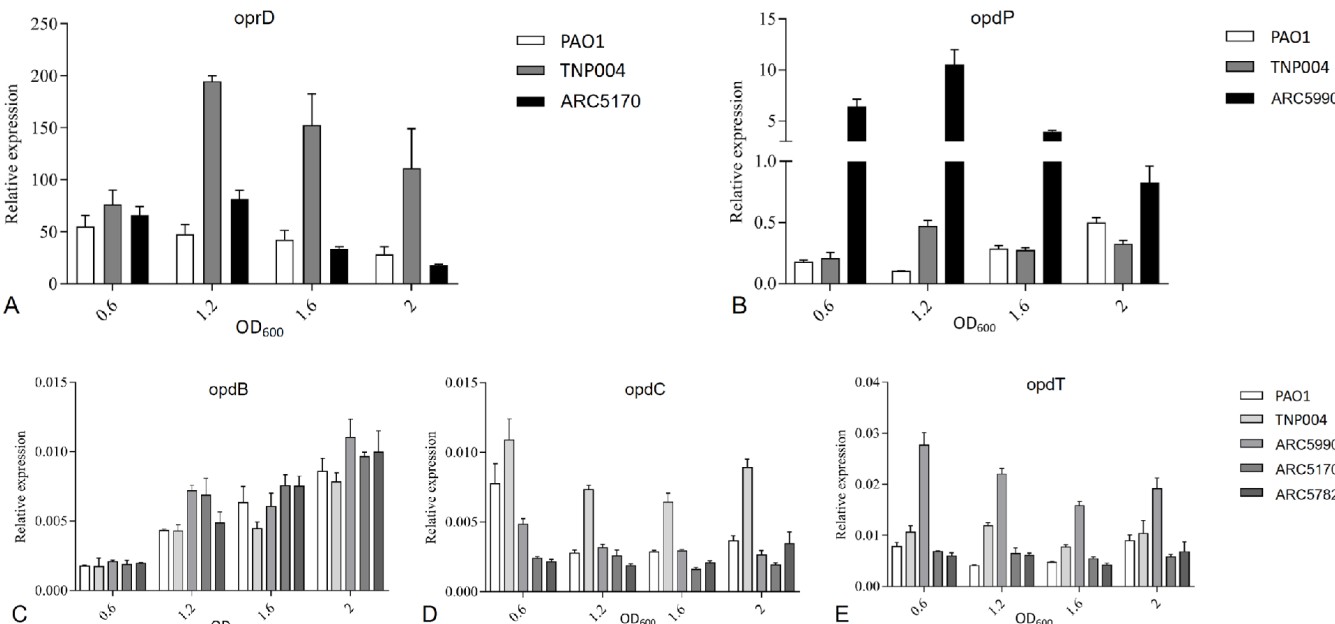

**FIG 4** Relative expressions of (A) oprD, (B) opdP, (C) opdB, (D) opdC, and (E) opdT mRNAs in *P. aeruginosa* PAOI and four porin(s) mutant strains; mRNAs were extracted at four different points of bacterial growth, reported in the x-axis as the absorbance at 600 nm, and the relative expression reported on the y-axis is the mean of the transcription of three indipendent reference genes (PA3340, gyrA, and cysG). Data were analyzed by two-way analysis of variance (ANOVA), followed by Bonferroni multiple comparison *post-test*.

exponential phase ($OD_{600}$ 0.6), might explain the differences found between carbapenem MICs and the permeability coefficients obtained in this study.

During our permeability measurements performed during the late exponential growth phase, as a consequence of the low physiological expression of OprD, the deletion of this porin does not produce a strong decrease in permeability for carbapenems, except for imipenem. In contrast, the MICs, whose determination involves the passage through the exponential phase in the overnight culture, show the effect of OprD's higher expression in the early growth phase.

Interestingly, in TNP004, *oprD* mRNA appeared to be upregulated, but this did not reflect in the insertion of a functional porin in the outer membrane, as demonstrated by Western blot analysis. The bacteria are, therefore, able to respond to the lack of the functional porin by overexpressing *oprD* mRNA transcription, suggesting that the expression of this porin is controlled by a precise regulatory mechanism.

We performed the same screening on *opdP* mRNA in *P. aeruginosa* PAO1, TNP004, and ARC5990 (PAO1ΔoprD), and the results are shown in Fig. 4B.

In *P. aeruginosa* PAO1 the expression of OpdP does not seem to be regulated by cell density, but, in contrast to OprD, its expression is slightly increased during the early stationary phase. Moreover, its expression is increased 20-fold when *oprD* mRNA is not expressed as in the case of ARC5990 (PAO1ΔoprD). In TNP004, in fact, the relative *opdP* expression remains similar to that in PAO1, probably due to the simultaneous over-transcription of *oprD* mRNA. This finding highlights how the absence of a porin can be compensated for, but it does not fully elucidate the mechanism that prevails in the TNP004 strain.

It is important to notice that the relative mRNA expression of *opdP* in ARC5990 (PAO1ΔoprD) is ten times less than that of *oprD* in ARC5170 (PAO1ΔopdP); nevertheless, the permeability coefficients for all carbapenems, except for imipenem, are similar in the two considered strains.

However, OpdP has been described to exhibit a 30-fold higher conductance than OprD (30), and this can explain how a lower expression of this porin allows the uptake of all carbapenems at the same rate, with the exception of imipenem.

We finally quantified the relative mRNA expression of the *oprB, oprC,* and *oprT* porin genes, previously chosen for their high relative expression in minimal medium (24), in *P. aeruginosa* PAO1, TNP004, ARC5990 (PAO1Δ*oprD*), ARC5170 (PAO1Δ*opdP*), and ARC5782 (PAO1Δ*oprD*, Δ*opdP*).

The results for these porins are reported in Fig. 4C through E, respectively, and show that all three porins have a basal expression level, while only that of *opdB* mRNA seems to be directly proportional to cell density. However, their low expression does not seem to be crucial for antibiotic uptake, as shown by the MICs and the permeability coefficient determinations.

We also quantified the relative expression of *oprD* mRNA in two strains selected during the multistep resistance experiment. We chose LG01, derived from *P. aeruginosa* PAO1, carrying a mutated *oprD* sequence, and LG03, derived from ARC5170 (PAO1Δ*opdP*) that possesses an intact *oprD,* and we compared the relative expression of the porin at $A_{600}$ = 1.6, using the reference genes mentioned above.

The results reported in Fig. 5 show an increased expression of *oprD* mRNA, similarly to TNP004, in LG01 when compared to the wild-type, most likely due to the synthesis of a mutated unstable porin.

Interestingly, *oprD* mRNA expression in LG03 appears to be tenfold downregulated compared to PAO1, probably as a consequence of the already mentioned mutations found in the *nalD* or *dsbS* genes.

Thus, the so far undocumented mechanism results in a downregulation of the porin (verified by qRT-PCR and Western blot), which causes carbapenem resistance highlighted by the MIC values.

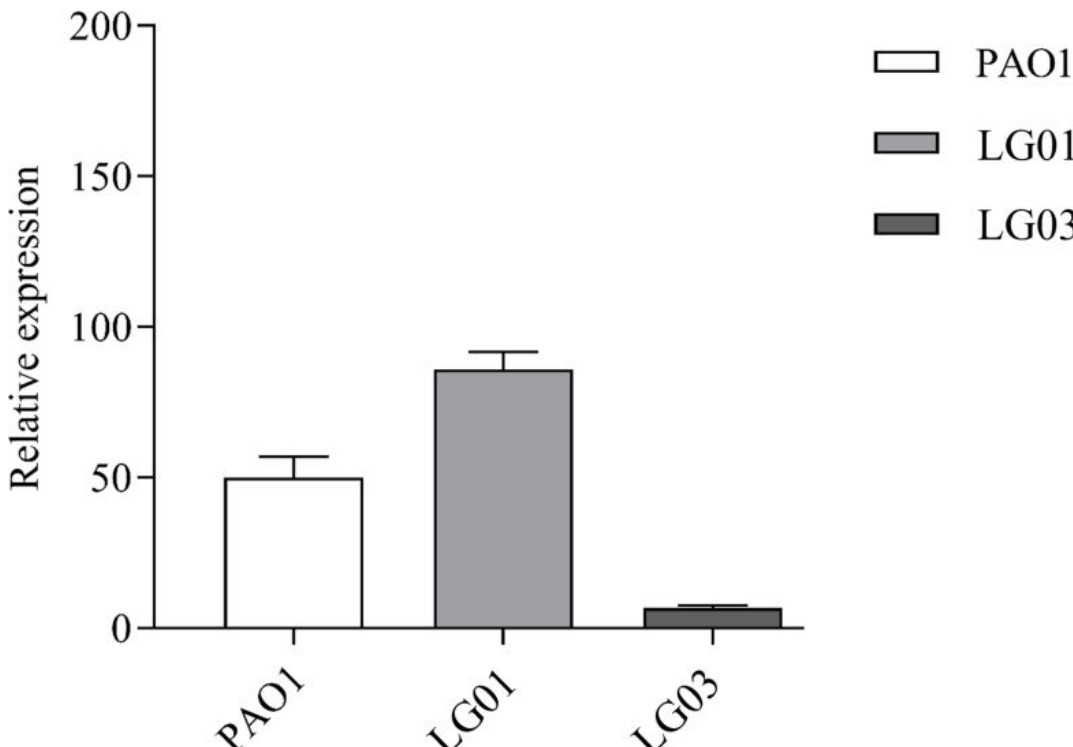

**FIG 5** Relative expression of oprD mRNA in *P. aeruginosa* PAOI, LG01, and LG03; mRNAS were extracted at A600 = 1.6, and the relative expression reported on the y axis is the mean of the transcriptions of three indipendent reference genes (PA3340, gyrA, and cysG) mentioned above. Data were analyzed by two-way analysis of variance (ANOVA), followed by Bonferroni multiple comparison *post-test*.

These experiments shed further light on OpdP, demonstrating that this porin is less expressed during the exponential growth phase, while its production increases when the bacterial culture enters the latent phase. This phenomenon turns out to be of particular importance during the permeability determinations, which were performed with cultures in the late exponential growth phase, but also during an *in vivo* infection. Conversely, the production of OprD follows an inverse pathway (Fig. 5A and B), and this mechanism contributes to explain why the MIC values appear to be mostly influenced by OprD. The determination of the MICs necessarily requires an exponential growth phase and, therefore, more affected by the presence/absence of the OprD porin in *P. aeruginosa*.

Besides this experimental evidence, it is, therefore, important to highlight how the simultaneous absence of OprD and OpdP porins might determine the poor efficacy of meropenem and biapenem during an antibiotic therapy, while the single deletion of OpdP can favor the appearance of a carbapenem resistance phenotype after exposure to meropenem in a concentration close to the MIC value.

Therefore, the presence of mutations/deletions of the OpdP porin should receive greater consideration from a clinical point of view, and further studies on the expression of this porin in clinical strains could lead to a better understanding of the resistance mechanisms mediated by this porin.

## MATERIALS AND METHODS

### Bacterial strains, growth conditions, plasmids, and antibiotics

Antibiotics were purchased from Sigma-Aldrich and nitrocefin from Oxoid Ltd. (Basingstoke, UK).

Biapenem was kindly provided by Dr. O. Lomovskaya from The Medicines Company (San Diego, CA, USA).

Bacterial strains are listed in Table 7, and the porin(s) deletion assessment is reported in supplemental data. Plasmids are reported in Table S1. Bacteria were grown aerobically at 37°C in LB, purchased from Sigma-Aldrich.

To perform planktonic cultures, a single colony was inoculated into liquid medium and incubated overnight (14–16 hours). Cultures were then diluted 1:20 into fresh media, and bacterial growth was monitored by following the absorbance at 600 nm using an Ultrospec 10 spectrophotometer (BioChrom, St Albans, UK).

**TABLE 7** Collection of *P. aeruginosa* strains used in the study[a]

| *P. aeruginosa* | Relevant characteristics | Reference |
|---|---|---|
| PAO1 | Wild-type | BCCM |
| PAO1-Jap | Presumptive TNP004 parental strain | (47) |
| TNP004 | ↓OprD | (47) |
| ARC545 | ARC presumptive parental strain | (24) |
| ARC5990 | Δ*oprD* | (24) |
| ARC5170 | Δ*opdP* | (24) |
| ARC5782 | Δ*oprD*, Δ*opdP* | (24) |
| ARC5998 | Δ*oprD*, Δ*opdC*, Δ*opdP*, Δ*opdT*, and Δ*opdB* | (24) |
| LG01 | *oprD* mutant, derived from PAO1 | This study |
| LG02 | *oprD* mutant, derived from ARC5170 | This study |
| LG03 | *oprD* mutant, derived from ARC5170 | This study |
| LG04 | *oprD* mutant, derived from ARC5170 | This study |
| LG05 | *oprD* mutant, derived from ARC5170 | This study |
| LG06 | *oprD* mutant, derived from ARC5170 | This study |
| LG07 | *oprD* mutant, derived from ARC5170 | This study |

[a]↓OprD indicates a downregulated expression of OprD.

The pKT240blaR shuttle vector (45) includes the *B. licheniformis blaR-CTD* gene under the control of the *lpp-lac* fusion promoter. The pKT240neg vector was derived from pKT240blaR where *blaR-CTD* was excised. Finally, pKT240blaR-gen was obtained by the addition of the *aac1* gene, in order to confer resistance to gentamicin. Both latter constructs were produced using the Gibson assembly technology (63), by amplifying fragments of the pKT240blaR plasmid and the *aac1* gene in the case of pKT240blaR-gen. Primers are listed in Table S2. The different plasmids were transformed in *E. coli* DH5α-Proteobacterium *Pseudomonas aeruginosa*. They were purified using the NucleoBond Xtra Maxi kit (Macharey-Nagel, Bethlehem, PA) and used to transform *P. aeruginosa* strains by electroporation using the Gene Pulser Xcell Electroporation System (Bio-Rad, Hercules, CA, USA), in an ice-cold 0.2 cm cuvette with the following parameters: 200 Ω, 25 μF, and 2.5 kV.

We further selected new *P. aeruginosa* mutant strains where the permeability of the outer membrane is affected. Those strains were obtained with the help of a multistep resistance experiment. Briefly, *P. aeruginosa* PAO1 and *P. aeruginosa* ARC5170 (PAO1Δ*opdP*), grown in liquid LB medium, were harvested in the mid- and late-exponential growth phase ($A_{600}$ ~1.2 and 1.6) and streaked ($10^9$ CFU) onto Petri plates containing two different meropenem concentrations (1 and 2 μg/mL), higher than the MICs determined for both parental strains (0.5 μg/mL). The plates were incubated for 16 hours at 37°C. Seven colonies (*P. aeruginosa* LG01, derived from *P. aeruginosa* PAO1, and *P. aeruginosa* LG02-LG07 derived from *P. aeruginosa* ARC5170) were then randomly selected for further characterizations.

## Susceptibility testing

Antimicrobial susceptibility was evaluated by broth microdilutions in cation-adjusted Mueller–Hinton broth (MHBII), purchased from Sigma-Aldrich, according to the Clinical and Laboratory Standards Institute guidelines (46). *P. aeruginosa* ATCC 27853 was used as a control strain, and data were collected in triplicate independent experiences.

## PCR amplification, DNA sequencing, and whole-genome sequencing (wgs)

PCR screenings on different targets were performed on crude extracts using OneTaq polymerase (New England Biolabs, Ipswich, MA, USA), while Q5 High-Fidelity enzyme (New England Biolabs) was used in case of a successive Sanger sequencing analysis.

The *oprD* sequence was determined by Sanger sequencing of PCR products, obtained using the pair of primers oprD_flankF and oprD_flankR (Table S2).

The nucleotide and protein sequences were analyzed using the blastn and blastp algorithms, available on the National Center of Biotechnology Information website (http://www.ncbi.nlm.nih.gov), and the alignments of the translated amino acid sequences of OprD were performed using the software Alignx (InforMax, Bethesda, MD, USA).

To perform whole-genome sequencing, genomic DNA was extracted using the NucleoSpin DNA Plus kit (Macherey-Nagel). Samples were then processed on a NovaSeq (Illumina, Inc., San Diego, CA, USA) sequencer, generating paired-end reads (2 × 150); raw reads were corrected by a homemade workflow, performing various steps of analysis, using software included in the BBTools package (64); briefly, reads were overlapped with BBMerge and subsequently quality-trimmed, and any remaining adapters were removed by the BBduk function. Tadpole and BBMap were in sequence used to perform a quick assembly, and BBduk was used for quality calibration; finally, BBNorm was used to normalize the coverage, and Tadpole was used for a final process of error-correction.

The sequence mapping was carried out with the Geneious (v10.2.6) software (65). Genome assembly was achieved by mapping single verified reads to a reference *P. aeruginosa* genome sequence (strain PAO1, GenBank accession number AE004091), and the resulting variations have been applied to the reference to generate the strain sequences.

Both Sanger and whole-genome sequencing were carried out at the GIGA-Genomics platform (GIGA-Genomics, Liège, Belgium).

## RNA extraction, cDNA synthesis, and quantitative qRT-PCR

Quantitative real-time PCR (qRT-PCR) was used to compare the expressions of porins genes, at four different moments of cellular growth ($A_{600} = 0.6$, 1.2, 1.6, and 2.0).

Total RNA isolation of the different aliquots was performed using the NucleoSpin RNA Plus kit (Macherey-Nagel), and rDNase (Macherey-Nagel) digestion was subsequently performed to eliminate any DNA trace. The samples were finally purified using the NucleoSpin RNA Clean-up XS kit (Macherey-Nagel) according to the manufacturer's recommendations.

RNA quantification was performed by measuring the absorbance at 260 nm with the help of a NanoVue spectrophotometer (GE Healthcare, Little Chalfont, UK). Then, 1 µg of RNA was retrotranscribed using the SuperScript III reverse transcriptase (Invitrogen, Waltham, MA, USA), triggered by random hexamers and supplemented with 0.4 U/µL of Ribosafe RNase Inhibitor (Bioline, USA). The reaction was carried out at 25°C for 5 minutes, 50°C for 60 minutes, and stopped by heating to 70°C for 15 minutes. The newly synthesized cDNA was diluted to 1:50 in RNase/DNase-free water and used as a target for qRT-PCR.

Amplifications were performed in 384-well plates with a QuantStudio 5 Real-Time PCR system (Thermo Fisher Scientific, Waltham, MA, USA) using Takyon Low Rox SYBR MasterMix Eurogentec (Seraing, Belgium); the list of primers used is reported in Table S3.

A control reaction was performed for each sample by using the original RNA mixture to verify the absence of residual DNA. Experiments were reproduced in four biological replicates and three technical replicates for each target gene.

The quality of the quantitative PCR was checked by the analysis of dissociation and amplification curves. For each primer pair, the mean reaction efficiencies were calculated using the LinRegPCR software (66) (Table S3). Those values were used to quantify relative gene expression levels by normalization using three reference genes (*PA3340*, *gyrA*, and *cysG*) with the qBase software (Biogazelle) (67). The reference genes were chosen because they were similarly expressed in the various growth phases. The adequacy of the reference genes to normalize gene expression in the experimental conditions was checked using the geNorm module in qBase (68).

## BlaR-CTD affinity

BlaR-CTD is the soluble C-terminal domain of the BlaR transmembrane protein that displays a high affinity for β-lactams characterized by the acylation constant ($k_2/K'$), the second-order rate constant for the formation of the acyl-enzyme adduct characterizing the acylation step efficiency. This rate constant can be determined by incubating a known concentration of the antibiotic, whose $k_2/K'$ is to be measured, together with a known concentration of a reporter antibiotic (r), whose kinetic parameters are known. The proportion of each acyl-enzyme formed at saturation depends on the concentration and $k_2/K'$ value of each antibiotic (equation 1) (69). Our reporter molecule was nitrocefin ($\Delta\varepsilon^{482} = 15{,}000\ \text{M}^{-1}\ \text{cm}^{-1}$).

The assay can be described as follows:

$$\frac{[EC^*]r}{[EC^*]} = \frac{\left(\frac{k_2}{K'}\right)r * [C]r}{\left(\frac{k_2}{K'}\right) * [C]} \text{ and } [EC^*]r + [EC^*] = [E_0]$$

$$\frac{[EC]_r}{[E_0] - [EC]_r} = \frac{(k_2/K')_r [C]_r}{(k_2/K')[C]} \qquad (1)$$

where [EC*]r and [EC*] are the concentrations of BlaR-nitrocefin and BlaR-antibiotic acyl-enzymes, respectively. [C]r and [C] correspond to the concentrations of nitrocefin

and of the tested antibiotic, respectively. $(k_2/K')r$ and $(k_2/K')$ represent the acylation rate constants of BlaR-CTD for the nitrocefin and the tested antibiotic.

We first determined the acylation rate constant of BlaR-CTD for nitrocefin. Purified BlaR-CTD was previously purified at the CIP (70). BlaR-CTD (20 µM), was added to a solution of nitrocefin (80 µM) containing increasing concentrations (25–600 µM) of ampicillin, whose $k_2/K'$ is known (1.3 $10^6$ M$^{-1}$· s$^{-1}$) (70). The variation of $A_{482}$ is directly proportional to the concentration of the BlaR-CTD-nitrocefin adduct. The $k_2/K'$ for nitrocefin was then calculated (3.6 $10^6$ M$^{-1}$· s$^{-1}$) and used as a competitor to determine the other affinity values. The acylation rate constants were so determined for cefalotin, ceftazidime, imipenem, meropenem, ertapenem, biapenem, and doripenem.

## β-Lactamase assays

The production of the class C AmpC β-lactamases was measured in crude cell extracts from the different *P. aeruginosa* PAO1 cultures as follows. The growth of the bacteria in LB at 37°C was monitored by measuring $A_{600}$. At a value of 1.6, the culture was divided into two 10 mL aliquots. One aliquot was used as a control. Antibiotic was added to the second aliquot at a final concentration equal to the maximum concentration of the antibiotic tested in the permeability assay (Table 8). The cultures were then incubated for a time corresponding to the incubation time with the selected antibiotic in the permeability test. One mL of each culture was then centrifuged at 13,000 *g* for 10 minutes. The pellet was washed twice and resuspended in 1 mL of 10 mM PBS buffer pH 7.4. Cells were lysed by sonication with the Bioruptor Plus Diagenode (Seraing, Belgium). The cellular extract was clarified by centrifugation at 13,000 *g* for 30 minutes at 4°C.

Positive controls for AmpC induction in *P. aeruginosa* PAO1 were analysed as previously described and incubated for six hours in presence of 50 µM ampicillin and 50 µM cefoxitin. The negative controls were made by the culture without antibiotic, grown for the different tested incubation times.

The protein concentration in each extract was measured with the help of a BCA protein assay kit (Pierce, Rockford, IL). The β-lactamase activity of the extract was determined by measuring the initial rate of hydrolysis of 100 µM nitrocefin. All the enzymatic assays were performed in 10 mM PBS buffer pH 7.4 at 30°C. The specific activity of the different samples was the rate of hydrolysis of each substrate expressed in nmoles per minute per milligram of protein.

TABLE 8  Different antibiotics concentrations tested during permeability experiments

| Antibiotics | Concentrations tested (µM) | | |
|---|---|---|---|
| Benzylpenicillin | 40 | 20 | 10 |
| Cefoxitin | 30 | 15 | 7.5 |
| Cefuroxime | 60 | 30 | 15 |
| Cefotaxime | 60 | 30 | 15 |
| Ampicillin | 20 | 10 | 5 |
| Cephaloridine | 8 | 4 | 2 |
| Imipenem | 0.02 | 0.01 | 0.005 |
| Imipenem[a] | 4 | 2 | 1 |
| Meropenem | 8 | 4 | 2 |
| Meropenem[b] | 20 | 10 | 5 |
| Ertapenem | 7.5 | 5 | 2.5 |
| Doripenem | 4 | 2 | 1 |
| Biapenem | 0.04 | 0.02 | 0.01 |
| Biapenem[b] | 4 | 2 | 1 |

[a]* refers to tests performed on strains lacking the OprD porin.
[b]while ^ refers to tests performed on strains deprived of both OprD and OpdP porins.

## Permeability determination

The antibiotic flux passing through the outer membrane can be described by the Flick's first law of flux (2).

$$J = -D \cdot A \cdot \frac{\Delta C}{\Delta x}, \tag{2}$$

where J is the flux of the antibiotic through the outer membrane OM, D the diffusion coefficient of the antibiotic, A the OM area (assumed as 132 cm$^2$), $\Delta C$ the concentration gradient of the antibiotic, and $\Delta x$ the OM thickness.

β-lactam flux in *P. aeruginosa* can be characterized by the permeability coefficient P (3), that is defined as the ratio between the diffusion coefficient and the OM thickness. It can be defined as:

$$P = -\frac{D}{\Delta x}. \tag{3}$$

The antibiotic flux is defined by equation 4:

$$J = P \cdot A \cdot \Delta C. \tag{4}$$

The estimation of the antibiotic flux can be achieved by expressing the high affinity BlaR-CTD in the bacterial periplasm, allowing the direct quantification of the β-lactam concentration present in the periplasmic space, and consequently to measure the permeability coefficient (5)

$$P = \frac{d(EC^*)/dt}{A \cdot [C_e]} \tag{5}$$

where EC* is the concentration of the BlaR-CTD-β-lactam adduct and [C$_e$] the external β-lactam concentration.

d(EC*)/dt is equal to the slope of the line reflecting the increase of the acyl-enzyme concentration *vs* the incubation time (Fig. 6B).

Antibiotic diffusion in the periplasmic space was analyzed in planktonic cultures. The different *P. aeruginosa* strains, previously transformed with pKT240blaR (or pKT40blaR-gen in the case of TNP004), were grown in LB medium added with selection antibiotic (50 µg/mL tetracycline or 10 µg/mL gentamicin, respectively). When the culture reached the late exponential phase (A$_{600}$ ≃ 1.6) a prefixed β-lactam concentration was added to the medium and aliquots (1 mL) were harvested at different incubation times. Two microliters of the metallo-β-lactamase VIM-2 (1 mg/mL) was added to the bacterial culture to hydrolyze all the antibiotics present outside the cells and consequently interrupt its permeation in an active form. EDTA (1 mM) was then added in order to chelate the metal ions and inactivate the metallo-β-lactamase. The crude extract was obtained by means of ten cycles of refrigerated sonication at 4°C, performed with the Bioruptor Plus Diagenode. The quantification of BlaR-CTD acylated by the tested antibiotic was performed by adding 2.5 µM Bocillin FL (Invitrogen), a fluorescent derivative of penicillin V, to the crude extracts. The proteins of the different cell extracts were separated by SDS-PAGE, and the fluorescence intensity of BlaR-CTD recorded using a Typhoon Trio +imager and Image Quant TL software (GE Healthcare) (Fig. 6A). The quantification of the BlaR-CTD-Bocillin adduct *vs* the incubation time in presence of an unlabeled β-lactam allowed the determination of the quantity of BlaR-CTD acylated with the unlabelled compound. Analyses were performed in duplicate at three different antibiotic concentrations as reported in Table 8.

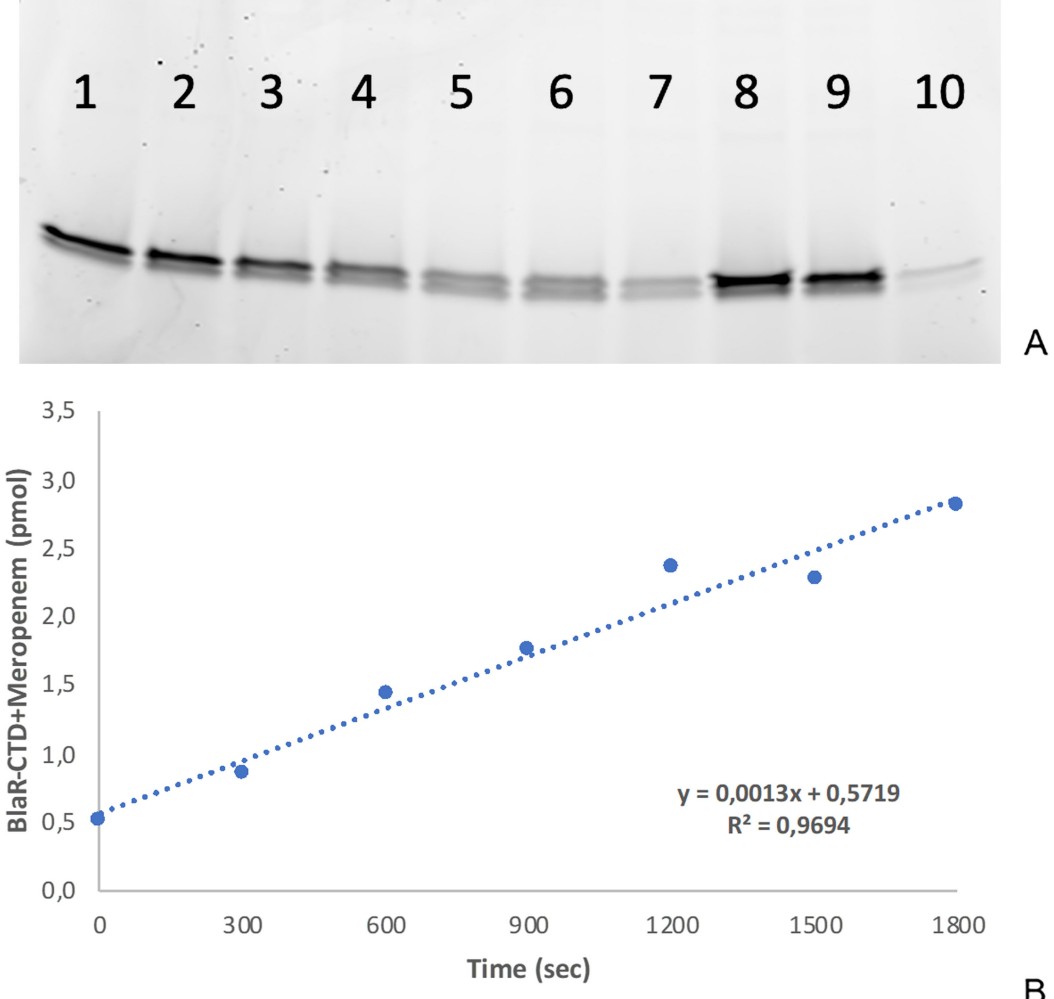

**FIG 6** (A) Densitometric analysis of the fluorescence signal of the BlaR-CTD-Bocillin adduct, shown as an example; samples from 1 to 7 represent different aliquots, taken every 5 minutes for determining meropenem uptake in ARC5782 (PAO1ΔoprD, ΔopdP); samples 8 and 9 are aliquots taken before the addition of meropenem to the culture, representing the total quantity of produced BlaR-CTD; sample 10 is a not sonicated aliquot, used to quantify possible BlaR-CTD release in the medium. (B) Graph representing the increase of the meropenem-BlaR-CTD adduct as a function of time; this quantification was made possible by subtracting the values obtained during the experiment (samples 1-7 of panel A) from the total BlaR-CTD produced in an aliquot (samples 8 or 9 of panel A). The slope of the line represents the antibiotic flux passing through the outer membran.

## Protein extraction, SDS, and Western Blot

The outer membrane profiling was performed on LB cultures as previously described (71).

SDS-PAGE was carried out according to the Laemmli protocol, using 4%–20% Mini-Protean (Bio-Rad) gels and Coomassie blue staining.

For Western blot analysis, proteins were electroblotted onto a PVDF membrane, using a commercial kit (Bio-Rad), and the protein marker V (pre-stained) VWR (Radnor, PA, USA) was used.

The membranes were then incubated with 1:2,000 rabbit-derived anti-OprD polyclonal antibody (72), kindly provided us by Dr Thilo Köhler (University of Geneva, Switzerland) and finally with the secondary antibody 1:5,000 goat anti-rabbit HRP antibody (Bio-Rad).

For the revelation, Clarity Western ECL substrate kit (Bio-Rad) was added, and signals were detected using an ImageQuant LAS 4000 camera (GE Healthcare).

## ACKNOWLEDGMENTS

We thank Prof. H. Yoneyama and Dr. A. Miller for the kind gifts of *P. aeruginosa* porin mutant strains.

We thank Dr O. Lomovskaya for the kind gift of biapenem.

Amisano F. was supported by the Excellence grants in WBI (Grant ID 2015/242329, Wallonie Bruxelles International) and the work was supported by Fund for Scientific Research (FRS-FNRS) Belgium.

## AUTHOR AFFILIATIONS

[1]InBioS, Center for Protein Engineering, Biological Macromolecules, Department of Life Sciences, University of Liège, Liège, Belgium
[2]InBioS - PhytoSystems, Functional Genomics and Plant Molecular Imaging and Centre for Assistance in Technology of Microscopy (CAREm), University of Liège, Liège, Belgium
[3]InBioS-PhytoSystems, Translational Plant Biology, University of Liège, Liège, Belgium

## AUTHOR ORCIDs

Francesco Amisano http://orcid.org/0000-0003-0822-4909
Moreno Galleni http://orcid.org/0000-0003-0992-0391

## FUNDING

| Funder | Grant(s) | Author(s) |
| --- | --- | --- |
| Wallonie Bruxelles International - WBI | 2015/242329 | Francesco Amisano |
| Fonds De La Recherche Scientifique - FNRS (FNRS) | J0081.20 | Moreno Galleni |

## AUTHOR CONTRIBUTIONS

Francesco Amisano, Data curation, Investigation, Methodology, Validation, Visualization, Writing – original draft | Paola Mercuri, Conceptualization, Formal analysis, Methodology, Project administration, Resources, Supervision, Validation, Visualization, Writing – original draft, Writing – review and editing | Steven Fanara, Conceptualization, Formal analysis, Investigation, Methodology, Writing – review and editing | Olivier Verlaine, Conceptualization, Data curation, Methodology, Software, Supervision, Validation, Writing – review and editing | Patrick Motte, Resources, Writing – review and editing | Jean Marie Frère, Conceptualization, Supervision, Writing – original draft, Writing – review and editing | Marc Hanikenne, Methodology, Supervision, Writing – original draft, Writing – review and editing | Moreno Galleni, Conceptualization, Data curation, Funding acquisition, Investigation, Methodology, Project administration, Resources, Supervision, Validation, Writing – original draft, Writing – review and editing

## DATA AVAILABILITY

The assembled genomic sequences of P. aeruginosa isolates were deposited under the Bioproject number PRJNA985251 in the NCBI database (https://www.ncbi.nlm.nih.gov/bioproject/)
Sanger sequencing data have been deposited on Genbank, under the accession numbers OR069747, OR069748, OR069749 and OR069750.

## ADDITIONAL FILES

The following material is available online.

Supplemental Material

**Fig. S1 (Spectrum00495-24-s0001.tiff).** Supplemental figure 1.

**Fig. S2 (Spectrum00495-24-s0002.tiff).** Supplemental figure 2.

**Fig. S3 (Spectrum00495-24-s0003.tiff).** Supplemental figure 3.

**Supplemental material (Spectrum00495-24-s0004.docx).** Porin(s) deletion assessment; RT-PCR reference genes.

**Table S1 (Spectrum00495-24-s0005.docx).** Collection of plasmids used in this study.

**Table S2 (Spectrum00495-24-s0006.docx).** Collection of primers used in this study for PCR amplification and sequencing.

**Table S3 (Spectrum00495-24-s0007.docx).** Sequences and reaction efficiencies of quantitative RT-PCR primer pairs.

## Open Peer Review

**PEER REVIEW HISTORY (review-history.pdf).** An accounting of the reviewer comments and feedback.

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
