## [Reviewer comments · Microbiology Spectrum]

Microbiology Spectrum

Outer membrane permeability of *Pseudomonas aeruginosa* through β -lactams: new evidence on the role of OprD and OpdP porins in antibiotic resistance

Francesco Amisano, Paola Sandra Mercuri, Steven Fanara, Olivier Verlaïne, Patrick Motte, Jean-Marie Frère, Marc Hanikenne, and Moreno Galleni

Corresponding Author(s): Moreno Galleni, Universite de Liege

Review Timeline:

Submission Date:	February 22, 2024
Editorial Decision:	April 30, 2024
Revision Received:	August 28, 2024
Accepted:	October 19, 2024

Editor: Mariagrazia Perilli

Reviewer(s): The reviewers have opted to remain anonymous.

Transaction Report:

DOI: <https://doi.org/10.1128/spectrum.00495-24>

Re: Spectrum00495-24 (Outer membrane permeability of *Pseudomonas aeruginosa*: elucidating the role of porins in antibiotic resistance)

Dear Prof. Moreno Galleni:

Thank you for the privilege of reviewing your work. Below you will find my comments, instructions from the Spectrum editorial office, and the reviewer comments.

Revision Guidelines

Sincerely,
Mariagrazia Perilli
Editor
Microbiology Spectrum

Reviewer #1 (Comments for the Author):

Resistance to carbapenems in non-fermenters such as *P. aeruginosa* is spreading worldwide, challenging the use of this last resort antibiotics. Carbapenem resistance in *P. aeruginosa* is multifactorial, and there are still several aspects incompletely understood.

This work by Amisano and coworkers address a fundamental issue in this regard, by studying the specific roles of porins OprD

and OpdP in this phenomenon. The authors exploit the high affinity of the penicillin-binding protein BlaR-CTD towards beta-lactams to quantify the levels of antibiotic going through the outer membrane, and they quantitate the expression of the different porins by RT-PCR. Overall, this is a highly relevant contribution to the field.

There are some (minor) aspects that the authors may want to consider to improve the quality of their work:

1. The title of the paper is more overarching than the paper content, and might be misleading for the reader, since the authors specifically address experimentally the expression and permeability of two specific porins. This is by no means a negative opinion on the work, but a more specific title would be more suitable.
2. The results of the whole genome sequencing is somehow confusing, since it seems like a preliminary study that may give rise to further mechanistic information. The authors themselves acknowledge this in the Discussion section. Maybe these results can be considered as preliminary, and explored later, since the rest of the article stands by itself.

Reviewer #2 (Comments for the Author):

This data-rich manuscript from a well-recognized research team contains useful information concerning outer membrane permeability in *Pseudomonas aeruginosa*. The role of the OpdP porin in carbapenem entry into *Pseudomonas* has been neglected in the past; this work highlights its importance. However, some of the methodology is not as rigorous as expected and needs to be better defined. Suggestions follow.

Specific comments

1. The authors may want to consider writing a "Results and Discussion" section rather than splitting them up. Discussion points are frequently inserted in the Results section.
2. Lines 246/590. The BlaR-CTD introduced into *Pseudomonas* originated from *Bacillus licheniformis*. It is unknown whether the insertion of a *Bacillus* "PBP" may have disturbed normal carbapenem uptake mechanisms in *Pseudomonas*. Can the authors make it clear in the Results section as to which studies supported the use of this beta-lactam sponge?
3. Tables 1 and 2. Please explain what the first column ("CLSI Standard") refers to.
4. Lines 252-254. Data in Table 2 exhibited more than 4-fold increases in MICs for the three cephalosporins and also for the non-beta-lactams tetracycline and gentamicin. The increases were especially high for the PA0509 mutant.
5. Table 3. Was a purified BlaR used for the acylation studies? More detail is needed as to the source of the enzyme.
6. Lines 255-257. The use of different antibiotics for different sets of experiments was sometimes confusing. How do the authors know there is low permeability to the three antibiotics? The permeability of the three beta-lactams was not provided in Table 5. And, the "high acylation rate of BlaR" in Table 3 is provided only for ceftazidime that has a lower acylation rate than any of the carbapenems (which did not exhibit MIC increases in the presence of BlaR). Please revisit this argument (that really belongs in the discussion).
7. Table 4. Please define what "/" means - negative control?
8. Line 266. The data in Table 4 does not provide the "concentration" of AmpC, but instead the "specific activity." These data would be more meaningful if drug concentrations were related to the K_m values for the substrates.
9. The data in Table 4 are unsatisfying. Most antibiotics were tested only once, and at different time points. Historical data demonstrate that imipenem should serve as an inducer, but it was tested only at a 12 minute time point. All drugs should have been tested at the same time points for a valid comparison. It's notable that cefoxitin induced AmpC at 40 and 360 min, but ampicillin did not exhibit induction at 30 min, although it induced at 360 min. The authors should provide additional explanations for the data.
10. Line 339-341. This conclusion regarding OprD and conductance needs more discussion.
11. Figure 5 can be omitted; the data can be included in the text.
12. Line 429 ff. This paragraph should include the names of the strains being discussed.
13. Line 476. It's not clear why the porin deletion assessment is in a separate supplement.
14. Lines 480-1. Were colony counts performed to determine the initial inoculum?
15. Line 495. It's not clear what is meant by "In synthesis".
16. Line 507. Were the triplicate MIC assays run on different days?
17. The authors should check their manuscript for single sentence paragraphs. Many of these can be combined (as in lines 350 to 361).

Responses to Reviewers

Reviewer #1 (Comments for the Author): **Answers highlighted in yellow**

Resistance to carbapenems in non-fermenters such as *P. aeruginosa* is spreading worldwide, challenging the use of this last resort antibiotics. Carbapenem resistance in *P. aeruginosa* is multifactorial, and there are still several aspects incompletely understood.

This work by Amisano and coworkers address a fundamental issue in this regard, by studying the specific roles of porins OprD and OpdP in this phenomenon. The authors exploit the high affinity of the penicillin-binding protein BlaR-CTD towards beta-lactams to quantify the levels of antibiotic going through the outer membrane, and they quantitate the expression of the different porins by RT-PCR. Overall, this is a highly relevant contribution to the field.

There are some (minor) aspects that the authors may want to consider to improve the quality of their work:

1. *The title of the paper is more overarching than the paper content, and might be misleading for the reader, since the authors specifically address experimentally the expression and permeability of two specific porins. This is by no means a negative opinion on the work, but a more specific title would be more suitable.*

We proposed a new title: **see line1-2**

**Outer membrane permeability of *Pseudomonas aeruginosa* through β -lactams:
New evidence on the role of OprD and OpdP porins in antibiotic resistance**

2. *The results of the whole genome sequencing is somehow confusing, since it seems like a preliminary study that may give rise to further mechanistic information. The authors themselves acknowledge this in the Discussion section. Maybe these results can be considered as preliminary, and explored later, since the rest of the article stands by itself.*

See lines 160-207. We removed the results concerning the mutations found in the two reference strains and we only mentioned the sequencing of TNP004 and LG03 in order to exclude the role of other polymorphisms in the resistance.

Reviewer #2 (Comments for the Author): **Answers highlighted in cyan**

This data-rich manuscript from a well-recognized research team contains useful information concerning outer membrane permeability in *Pseudomonas aeruginosa*. The role of the OpdP porin in carbapenem entry into *Pseudomonas* has been neglected in the past; this work highlights its importance. However, some of the methodology is not as rigorous as expected and needs to be better defined. Suggestions follow.

Specific comments

1. The authors may want to consider writing a "Results and Discussion" section rather than splitting them up. Discussion points are frequently inserted in the Results section.

We have followed the suggestion of the reviewer 2. We merged results and discussion. See lines 138-397 highlighted in yellow and cyan

2. Lines 246/590. The BlaR-CTD introduced into *Pseudomonas* originated from *Bacillus licheniformis*. It is unknown whether the insertion of a *Bacillus* "PBP" may have disturbed normal carbapenem uptake mechanisms in *Pseudomonas*. Can the authors make it clear in the Results section as to which studies supported the use of this beta-lactam sponge?

See line 236-264, table 2 and Reference 45.

The use of a beta-lactam sponge, as reported in the text, was firstly proposed by Lakaye and coworkers (Lakaye et al., 2002) as a new method to determine the outer membrane permeability to beta-lactams.

It was verified by MIC determination that the presence of the *Bacillus* derived BlaR-CTD in the periplasm of *P. aeruginosa* does not alter carbapenems' MIC (lines 248-250). Moreover, even if any disturbing effect could be present, it would affect all the studied strains, not involving the presented synergic role of OprD and OpdP porins in the internalization of meropenem and biapenem.

Finally, the authors believe that it is impossible to have a method that could be 100% exempt from minimal disturbances on the physiological bacterial growth and that the method used in this study represent a good compromise for these measures.

3. Tables 1 and 2. Please explain what the first column ("CLSI Standard") refers to.

See legends of table 1 and 2.

CLSI standard refers to acceptable limits for quality control strains used to monitor accuracy of MICs. CLSI susc. refers to the MIC susceptibility breakpoints interpreted by CLSI (60).

NA: not available, ND not determined

We added this description in the legend, and we thought to add a column reporting the CLSI sensibility breakpoints to compare the obtained results.

We changed the CLSI reference to the 30th edition to include CLSI standard value for biapenem.

4. Lines 252-254. Data in Table 2 exhibited more than 4-fold increases in MICs for the three cephalosporins and also for the non-beta-lactams tetracycline and gentamicin. The increases were especially high for the PA0509 mutant.

Gentamicin and tetracycline, as reported in lines 638-639, are the resistance determinants carried by the plasmids used to transform TNP004 and all the other used strains respectively. The increase of the two MICs were reported only for strains transformed with the two plasmids carrying the gentamicin and tetracycline resistance factors.

PA0509, deprived of the major efflux pumps was removed from the text during the drafting phase and its MIC value are in accordance to the lack of different resistance determinants. However, it was forgotten in table 2 and so we have removed it and we added MICs results obtained for the strain transformed with the empty vector pKT240neg (without *blaR-CTD*).

The major difference we found are corresponding to 4 serial dilutions. We used 2- to 4- fold and we changed at line 253/ 2- to 4 -fold dilutions

5. Table 3. Was a purified BlaR used for the acylation studies? More detail is needed as to the source of the enzyme.

See lines 514-515.

We used a purified BlaR-CTD available in our laboratory (Duval et al., 2003).

6. Lines 255-257. The use of different antibiotics for different sets of experiments was sometimes confusing. How do the authors know there is low permeability to the three antibiotics? The permeability of the three beta-lactams was not provided in Table 5. And, the "high acylation rate of BlaR" in Table 3 is provided only for ceftazidime that has a lower acylation rate than any of the carbapenems (which did not exhibit MIC increases in the presence of BlaR). Please revisit this argument (that really belongs in the discussion).

See lines 249-260 of the revised manuscript

....A similar conclusion was made by Montaner and coworkers studying the effect of AmpC hyperexpression (Montaner et al., Microbiology Spectrum.2022. 11(1)e003038). The periplasmic BlaR-CTD production, similarly to AmpC hydrolysis, could reduce the intracellular concentration of these three antibiotics to a limit level that would change the PBPs occupancy causing increases in MICs or otherwise could induce AmpC expression. However, whatever the exact cause is, we excluded these antibiotics from our permeability study, preferring to concentrate on antibiotics whose MICs were not altered by BlaR-CTD expression.

7. Table 4. Please define what "/" means - negative control?

See lines 539-540 of the revised manuscript

The slash (/) stands for "negative control", a culture where no antibiotic was added and where the AmpC induction was measured to compare with the cultures in which antibiotic was added (as specified in line 609 of the first version of the manuscript which correspond to lines 539 and 540 of the revised version).

In addition, we provided the description for the slash in the **legend of table 4:**

"Periplasmic AmpC concentration and specific activity of the AmpC β -lactamase in the presence or absence (/) of β -lactams for the different *P. aeruginosa* cultures. The activity was followed by nitrocefin hydrolysis. (/) refers to a negative culture where no antibiotic was added."

8. Line 266. The data in Table 4 does not provide the "concentration" of AmpC, but instead the "specific activity." These data would be more meaningful if drug concentrations were related to the K_m values for the substrates.

See table 4.

We modified table 4 adding the periplasmic AmpC concentration determined after exposure to different antibiotics.

9. *The data in Table 4 are unsatisfying. Most antibiotics were tested only once, and at different time points. Historical data demonstrate that imipenem should serve as an inducer, but it was tested only at a 12 minute time point. All drugs should have been tested at the same time points for a valid comparison. It's notable that cefoxitin induced AmpC at 40 and 360 min, but ampicillin did not exhibit induction at 30 min, although it induced at 360 min. The authors should provide additional explanations for the data.*

We would like to comment the reviewer concern.

We knew from the literature that AmpC is inducible and some antibiotics (i.e. ampicillin, imipenem, cefoxitin) are good inducers.

However, all the beta-lactams may be AmpC inducers, and we wanted to verify that this event was not a cause of variability during the assays we performed.

For this reason, we used positive controls only to verify the robustness of our method for the determination of AmpC induction.

Later, we only focused on the antibiotics tested during our project, at the maximal concentration tested for permeability determination and for an incubation time that corresponded to the maximum time of incubation performed during the experience of outer membrane permeability determination.

We were not interested to assess, for example, the time or the concentration taken to induce AmpC in the presence of imipenem, rather to demonstrate that at the low concentration tested and in the 12 minutes time frame where the permeability measure was performed for imipenem the level of AmpC induction were comparable to a culture where no antibiotic was added

10. *Line 339-341. This conclusion regarding OprD and conductance needs more discussion.*

We reported the wrong reference. It is not "Liu et al 2022" but "Liu et al 2012".

It has been changed in the bibliography section

Reference 30

30. Liu J, Wolfe AJ, Eren E, Vijayaraghavan J, Indic M, van den Berg B, Movileanu L. 2012. Cation selectivity is a conserved feature in the OccD subfamily of Pseudomonas aeruginosa. *Biochim Biophys Acta*. 1818:2908-2916.

The modifications correspond to lines 356-357.

We changed the sentence "In the article OpdP conductance is reported to be 667 ± 71 pS, compared to 21 ± 3 pS for OprD" by "However, OpdP has been described to exhibit a 30-fold higher conductance than OprD (30), and this can explain how a lower expression of this porin allows the uptake of carbapenems at the same rate, with the exception of imipenem".

11. *Figure 5 can be omitted; the data can be included in the text.*

We believe that figure 5 shows an important result for our study (the OprD downregulation in a mutant selected during our study and the absence of a known resistance mechanism that could explain this mechanism).

This result is already mentioned in the text (lines 376-377). We modified the text as follow:

Interestingly, *oprD* mRNA expression in LG03 appears to be 10-fold downregulated compared to PAO1, probably as a consequence of the already mentioned mutations found in *nalD* or *dsbS* genes.

12. Line 429 ff. This paragraph should include the names of the strains being discussed.

We followed the suggestion of the referee.

13. Line 476. It's not clear why the porin deletion assessment is in a separate supplement.

Comments.

We received different strains already described in published articles for specific porin(s) deletions.

We verified the absence of these porins by PCR and we believe that this result is useful to verify the correctness of our results but not essential for the text that is already rich in different results.

For this reason, we believe that this section should be placed in the supplemental section.

14. Lines 480-1. Were colony counts performed to determine the initial inoculum?

Lines 281-282

No, we did not perform a colony count on the initial inoculum.

We tried to standardize the outer membrane permeability measures by starting from cultures in the late exponential phase, measuring the OD₆₀₀ and using the same value (1.6, lines 276-277, new lines 281-282).

The time needed to reach this value was around 6 to 7 hours and we believe that was sufficient to level any variability of the pre-inoculum.

15. Line 495. It's not clear what is meant by "In synthesis".

See Line 422 new manuscript version

We used in synthesis as a synonym for briefly

We have changed in the text with "Briefly" (new line 422).

16. Line 507. Were the triplicate MIC assays run on different days?

Comments and see line 507.

Yes, MICs were run on different days and even weeks and months. We added in the text at line 507 "triplicate independent experiences"

17. The authors should check their manuscript for single sentence paragraphs. Many of these can be combined (as in lines 350 to 361).

We modified the single sentences as suggested by the referee.

Further modifications

- 1) We removed the antibiotics (cefalotin, and ceftazidime) from table 3 since we were not able to obtain reliable coefficients for these two antibiotics.**
- 2) We also removed Ceftazidime, Cefepime and Piperacillin from table 4 since we didn't tested these antibiotics for the determination of the outer membrane permeability.**

Re: Spectrum00495-24R1 (Outer membrane permeability of *Pseudomonas aeruginosa* through β -lactams: new evidence on the role of OprD and OpdP porins in antibiotic resistance)

Dear Prof. Moreno Galleni:

thank you the revisions you made, now your manuscript has been accepted, and I am forwarding it to the ASM production staff for publication. Your paper will first be checked to make sure all elements meet the technical requirements. ASM staff will contact you if anything needs to be revised before copyediting and production can begin. Otherwise, you will be notified when your proofs are ready to be viewed.

Sincerely,
Mariagrazia Perilli
Editor
Microbiology Spectrum